# Metabolic Responses of the Microalga *Neochloris oleoabundans* to Extracellular Self- and Nonself-DNA

**DOI:** 10.3390/ijms241814172

**Published:** 2023-09-16

**Authors:** Mónica A. Zárate-López, Elizabeth Quintana-Rodríguez, Domancar Orona-Tamayo, Víctor Aguilar-Hernández, Jesús A. Araujo-León, Ligia Brito-Argáez, Jorge Molina-Torres, José Luis Hernández-Flores, Víctor M. Loyola-Vargas, Nancy E. Lozoya-Pérez, Edmundo Lozoya-Gloria

**Affiliations:** 1Centro de Investigación y de Estudios Avanzados del IPN (CINVESTAV-IPN), Unidad Irapuato, Km 9.6 Carretera Irapuato-León, Irapuato 36824, Guanajuato, Mexico; monica.zarate@cinvestav.mx (M.A.Z.-L.); jmolinat@cinvestav.mx (J.M.-T.); jose.hernandezf@cinvestav.mx (J.L.H.-F.); 2Centro de Innovación Aplicada en Tecnologías Competitivas (CIATEC), Omega # 201 Col. Industrial Delta, León 37545, Guanajuato, Mexico; dorona@ciatec.mx (D.O.-T.); nelppat@hotmail.com (N.E.L.-P.); 3Centro de Investigación Científica de Yucatán, A.C. (CICY), Calle 43 # 130, Chuburná de Hidalgo, Mérida 97205, Yucatán, Mexico; victor.aguilar@cicy.mx (V.A.-H.); alfredoaraujo@gmail.com (J.A.A.-L.); lbrito@cicy.mx (L.B.-A.); vmloyola@cicy.mx (V.M.L.-V.)

**Keywords:** lipids, MeJA, *Neochloris oleoabundans*, nonself-exDNA, peroxidase, phytohormones, polyphenols, ROS, self-exDNA, triglycerides

## Abstract

Stressed organisms identify intracellular molecules released from damaged cells due to trauma or pathogen infection as components of the innate immune response. These molecules called DAMPs (Damage-Associated Molecular Patterns) are extracellular ATP, sugars, and extracellular DNA, among others. Animals and plants can recognize their own DNA applied externally (self-exDNA) as a DAMP with a high degree of specificity. However, little is known about the microalgae responses to damage when exposed to DAMPs and specifically to self-exDNAs. Here we compared the response of the oilseed microalgae *Neochloris oleoabundans* to self-exDNA, with the stress responses elicited by nonself-exDNA, methyl jasmonate (MeJA) and sodium bicarbonate (NaHCO_3_). We analyzed the peroxidase enzyme activity related to the production of reactive oxygen species (ROS), as well as the production of polyphenols, lipids, triacylglycerols, and phytohormones. After 5 min of addition, self-exDNA induced peroxidase enzyme activity higher than the other elicitors. Polyphenols and lipids were increased by self-exDNA at 48 and 24 h, respectively. Triacylglycerols were increased with all elicitors from addition and up to 48 h, except with nonself-exDNA. Regarding phytohormones, self-exDNA and MeJA increased gibberellic acid, isopentenyladenine, and benzylaminopurine at 24 h. Results show that *Neochloris oleoabundans* have self-exDNA specific responses.

## 1. Introduction

Life on Earth is constantly exposed to danger, and plants have had to face infections by pathogens and herbivores for millions of years that inevitably cause them harm. Pathogen-Associated Molecular Patterns (PAMPs) are elicitors, such as flagellin and other microbial molecules, that are recognized by the plant’s immune system early [1]_._ Herbivore-Associated Molecular Patterns (HAMPs) are associated with responses mediated by hormones such as salicylic acid (SA) and jasmonic acid (JA) [2]. However, all these lesions release cell content that will be broken down from larger to simpler molecules by lytic enzymes. All these components of endogenous origin express cell damage and can therefore act as Damage-Associated Molecular Patterns (DAMPs), which are the oldest and possibly most conserved inducers of innate immunity in plants [3]. Those molecules act as endogenous danger messengers, triggering a series of chemical responses that protect the cell from harm or death. DAMPs have a variety of responses, ranging from early signaling mediated by Ca^2+^ cascades and activation of mitogen-activated protein kinases MAPKs, to the formation of reactive oxygen species (ROS) [4]. To date, some DAMPs belonging to nucleotides such as extracellular adenosine triphosphate (ATPe) and extracellular deoxyribonucleic acid (exDNA), have been identified; however, DAMPs associated with carbohydrates, proteins, and small peptides have also been found [3]. Exogenous PAMP and HAMP signaling molecules are responsible for activating the “nonself recognition” pathway [5,6,7]. In contrast, the perception of DAMPs activates the “damaged-self recognition” pathway through ROS, membrane depolarization, calcium fluxes, MAPKs cascade, and genetic activation [8,9].

It is known that fragmented extracellular self-DNA (self-exDNA) has an inhibitory effect on plant growth, which is impacted in a concentration-dependent and species-specific way, resulting in negative plant–soil feedbacks [10,11]. Currently, there is experimental evidence supporting self-exDNA as a DAMP that can provide species specificity in plant damaged-self recognition [12,13]. Responses related to the presence of DAMPs or stress in algae have already been studied. A cell extract of the macroalgae *Ascophyllum nodosum* has been shown to control the growth of the microalgae *Chlorella vulgaris*. At concentrations greater than 1% of the cell extract, the microalgae showed more than 80% growth inhibition and the antioxidant activity significantly decreased [14]. The report on macroalgae revealed that the components in the cell extracts, such as ATP, DNA, proteins, carbohydrates, etc., interact with microalgae cells, generating a response in their antioxidant defense mechanism. The first evidence on extracellular nucleotides participation as stimulators was in the macroalgae *Dasycladus vermicularis*, using fluorescent indicator dyes [15]. In that work, oxidative responses were observed upon physical injury and compared with uninjured cells induced with ATP analogues such as adenosine-5′-o-(3-thio-triphosphate). An inhibitor of ATP-dependent enzyme systems blocked the effects produced by ATPs and the inhibition of the oxidation effect. On the other hand, the production of ROS in the colonial microalga *Botryococcus braunii* was evaluated by fluorescence microscopy using CellROX Green after stress induction with NaCl, acetic acid, MeJA, sodium bicarbonate, and SA. All inducers generated fluorescence due to oxidative stress, but MeJA was more evident [16]. These results indicated that this microalga responds to stress through similar mechanisms to those of terrestrial plants. A recent study analyzed the impact of self-exDNA on the growth of the freshwater microalgae *Chlamydomonas reinhardtii* and the marine microalgae *Nanochloropsis gaditana*. The concentration of 30 ng µL^−1^ self-exDNA caused a reduction in cell density, in addition to an altered cell morphology with a palmelloid-type phenotype forming aggregates. No changes occurred with the external application of nonself-exDNA from *Sardina pilchardus* [17]. 

*Neochloris oleoabundans*, a terrestrial, oily green, unicellular microalgae, is spherical in shape and 3 to 6 mm in diameter and belongs to the *Chlorophyta* phylum [18]. This species was isolated from the sand dunes of Saudi Arabia, demonstrating a high adaptability to the extremely harsh environment. The cultivation of this microalgae is possible in both freshwater and marine water conditions [19]. Due to its high growth rate and adaptability, *N. oleoabundans* biomass has a wide range of applications, including the production of bioplastics, cosmetics, nutraceuticals, and vegetable oils production, among others [20]. The characteristic of these microalgae, which has been of interest to the scientific community and some industrial sectors, is its ability to accumulate up to 54% of its dry weight in total lipids under nitrogen deficiency [21]. Because of this property, it has been considered a study model for biodiesel production [20]. The main intracellular lipid components of these microalgae are C16:0, C18:0, C18:1, C18:2, and C18:3 type. Palmitic acid (C16:0) and stearic acid (C18:0) are both present in the intercellular space and in the cell wall, the others are components of the cell wall [22]. The accumulation of fatty acids suggests that *N. oleoabundans* has a high rate of lipid biosynthesis, perhaps as a response to increased abiotic stress caused by extreme environments, such as nitrogen deficiency, extreme temperatures, light changes, salinity, and nutrient deficiency, which all cause a physiological defense response [23,24]. In the case of *N. oleoabundans*, there is no information on the effects of DAMPs on lipid biosynthesis so far. In this work, we studied the response of these oleaginous microalgae when exposed to self-exDNA in comparison to nonself-exDNA from salmon sperm, and other stress factors such as MeJA and sodium bicarbonate. Responses to the different types of stress were analyzed by measuring peroxidase enzyme activity, polyphenols, lipids, and phytohormones production.

## 2. Results

### 2.1. Assay of the N. oleabundans Peroxidase Enzyme Activity after Treatments

The dose–response curve of the peroxidase enzyme activity of *Neochloris oleoabundans* indicated that 20 ng μL^−1^ was the optimal concentration of self-exDNA at which the first statistically significant response was observed Appendix A. As shown in this figure, after 15 min of self-exDNA application, the peroxidase enzyme activity was statistically significant at 20 and 30 ng μL ^−1^. However, at later times (30, 60, and 120 min), the enzyme activity decreased at the concentration of 30 ng μL ^−1^, but at 20 ng μL ^−1^, it was still increasing, being the highest at 60 min. According to these results, *N. oleoabundans* inductions were performed with self-ex and nonself-exDNA at 20 ng μL^−1^. Figure 1 shows the results of *N. oleabundans* peroxidase enzyme activity at different times after the addition of the four elicitors: self-exDNA, nonself-exDNA, MeJA, and NaHCO_3_, and without any elicitor. Treatment with self-exDNA resulted in a significant response after 5 min of exposure, then there were variations along the time, but they increased again at 120 min after exposure. Nonself-exDNA treatment showed no change and remained as a control along all assayed times. Treatment with NaHCO_3_ increased after 5 min, reaching the maximum at 15 min, and then decreasing later. MeJA treatment also increased continuously from 5 to 30 min, decreasing at 60 min, but reaching the maximum at 120 min. At 5 min, self-exDNA showed the highest enzyme activity induction over all other elicitors which indicated a fast and specific response. From 10 min to later, NaHCO_3_ and MeJA were the best elicitors compared to self-exDNA, but the enzyme activity induced by this last treatment was still at least half of the other two elicitors. 

### 2.2. Effect of Elicitors on the Production of Total Polyphenols in N. oleoabundans

Figure 2 shows the results of the total polyphenols of *N. oleabundans* after treatments. There were no significant changes for the first 24 h, although MeJA and NaHCO_3_ started increasing total polyphenols at 24 h. At 48 h, self-exDNA induced the polyphenols production by 32.6%, MeJA by 48.8%, and NaHCO_3_ increased to 67.4% of total polyphenol accumulation. The nonself-exDNA effect behaved like the control along all assayed times. Although total polyphenols were detected from the beginning in all cases, a clear increase was observed at 48 h mainly for NaHCO_3_ and MeJA. However, the effect of self-exDNA was also statistically significant at that time.

### 2.3. Effect of Inducers on the Production of Total Lipids in N. oleoabundans

The production of lipids is a specific response of the microalga *N. oleoabundans* to stress in general [24]. The application of self-exDNA resulted in significant lipid production at 24 and 48 h after induction as shown in Figure 3. At 0 h, all treatments had on average the same amount of lipids. At 24 h, lipid production was induced with self-exDNA, a 34.7% increase over control, and production was higher than all other treatments. At 48 h, self-exDNA treatment produced the most lipids; however, the other treatments also increased the production of lipids.

### 2.4. Effects of Elicitors on Triacylglycerols Production in N. oleoabundans

Lipids were extracted from *N. oleoabundans* after all treatments and triacylglycerols were analyzed by HPTLC at 0, 24, and 48 h, as shown in Figure 4. Samples were compared with standards of trilinolenin, trilinolein, and triolein [25]. The results showed that the triglyceride profile of the microalgae treated with self-exDNA, MeJA, and NaHCO_3_ showed a similar triacylglycerols profile, with more intense bands with Rf 0.28 to Rf 0.35 between trilinolenin and trilinolein. The intensity of these bands was similar with MeJA at all times, with NaHCO_3_ decreasing from 0 to 24 h, but then increasing at 48 h. Self-exDNA showed similar results at 0 and 24 h, but decreased at 48 h. Interestingly, very few triacylglycerols were detected at all times with nonself-exDNA, and the control samples contained triacylglycerols similar to nonself-exDNA at 0 and 24 h, but higher bands (Rf 0.35–0.45) were observed at 48 h. All treatments except nonself-exDNA induced the production of more desaturated triacylglycerols. In control samples, more saturated triacylglycerols were observed at 48 h, and nonself-exDNA seems to inhibit the production of triacylglycerols.

### 2.5. Effect of Elicitors on the Production of Phytohormones in Neochloris oleoabundans by UHPLC-MS

In this study, we analyzed the presence and endogenous response to stress of eight phytohormones: trans-zeatin and cis-zeatin (tZ/cZ), N6-isopentyladenine (IsoA), N6-benzylaminopurine (BAP), 3-indole acetic acid (IAA), methyl jasmonate (MeJA), salicylic acid (SA), abscisic acid (ABA), and gibberellin (GA3) in the microalgae *N. oleoabundans*. The effect of the elicitors on the production of these phytohormones along the time is shown in Figure 5. Phytohormones tZ/cZ and ABA were not detected in the microalgae *N. oleoabundans* at any time or treatment. All other phytohormones were present at 0 h with all treatments, with the exception of the GA (Figure 5d) that was within the detection limit with the MeJA treatment, suggesting sample degradation. Only IsoA, BAP, and GA (Figure 5b–d) showed statistical differences with self-exDNA at 24 h. IsoA and GA were statistically higher with self-exDNA and MeJA, and BAP was only significant with self-exDNA. In contrast, SA production was lower with self-exDNA and other treatments including nonself-exDNA, compared to the control. The SA was higher in all treatments at zero time, but it decreased at 24 and 48 h after treatments, except for the control at 24 h, where it stayed at the same concentration as zero time. These results suggest that all treatments decreased SA concentration at 24 h and later. IAA and MeJA (Figure 5a,e) were at similar concentrations in all samples at 24 and 48 h. At 48 h, any treatment induced the production of any phytohormone, and NaHCO_3_ had no effect on any phytohormone at all assayed times.

## 3. Discussion

ROS are at the basal level in all cells as they act as signaling molecules. However, under biotic or abiotic stress conditions, their values rise and cause oxidative damage degrading biomolecules such as proteins, DNA, lipids, and carbohydrates that lead to cell death [26]. The optimal concentration of self-exDNA Appendix A to get the first significant response in *N. oleoabundans* was 20 ng μL^−1^, similar to that reported [17] where 10 ng μL^−1^ had no effect in the cell density of *Chlamydomonas reinhardtii* and *Nannochloropsis gaditana*, but 30 ng mL^−1^ caused a significantly reduced cell density and aggregates in both microalgae. In that aspect, self-exDNA at optimal concentration induced *N. oleoabundans* peroxidase activity after 5 min of exposure (Figure 1), then went down at 10 min and up again at 30 min, down at 60, and reached the maximum at 120 min, suggesting that H_2_O_2_ and hence ROS were produced three times during this time interval. It is interesting that this response is fast and statistically higher than control and nonself-exDNA, along all the time up to 120 min. Therefore, oxidative stress is constant throughout all assayed times. MeJA treatment showed a similar response to self-exDNA, producing a rapid increase in peroxidase activity response until 30 min, decreasing at 60 min, and then increasing to its maximum at 120 min in a biphasic behavior. We observed that the oxidative stress was again constant throughout all assayed times. The cell uses antioxidant enzymes to minimize ROS and decrease the oxidative burst. In that aspect, peroxidase activity responds in a different way to the NaHCO_3_ treatment, showing a fast increase from 5 to 15 min and a decrease to levels similar to control at 120 min. As far as we know, this is the first report of peroxidase enzyme activity at earlier times in microalgae. Other peroxidases and/or H_2_O_2_ production have been studied after MeJA [16,27,28] and NaHCO_3_ [16,29] treatments in several microalgae, but at longer times of treatment and higher MeJA and NaHCO_3_ concentrations. In all cases, ROS or peroxidase enzyme activities increased after treatments.

Polyphenols have been reported for several species of microalgae, correlated with their antioxidant potential [30,31,32,33]. In that aspect, the microalgae *N. oleoabundans* were reported to produce a high concentration of polyphenols, that is 9.8 mg per gram of dry biomass and total phenol content (TPC) averaging 6 mg [34]. Phenolics have been associated with antioxidant protective effects in *Spirulina maxima*, *Tetradesmus obliquus*, and *Chlorella* ssp. [35,36]. Those data are similar to the amount of TPC obtained in this work since the values fall within this range as shown in Figure 2. At 48 h after treatment, self-exDNA, MeJA, and NaHCO_3_ statistically increase the TPC content. 

It is well known that nitrogen deficiency induces a significant increase in the lipid content in many species of microalgae. All assayed species synthesized C14:0, C16.0, C18:1, C18:2, and C18:3 fatty acids [37]. Other factors such as salt, iron, light intensity, and temperature have been studied regarding their effects on lipids production [38,39,40]. As mentioned before, one of the most interesting species is *N. oleoabundans*, which under nitrogen deficiency, typically yields 35–54% lipids of cell dry weight. Triacylglycerols comprised 80% of the total lipids, and saturated, monounsaturated, and di-unsaturated octodecanoic acid represented approximately one-half of the total fatty acids [21,41]. It has been proposed that cells accumulate large quantities of chlorophyll molecules when the nitrogen source is abundantly available. Then, when the external nitrogen sources are exhausted, cells begin to utilize chlorophylls as an intracellular nitrogen source [42]. It has been reported that antioxidant enzymes and their isoforms are involved in one of the mechanisms operating during N starvation and recovery in *P. cruentum*, *T. minutus*, and *S. incrassatulus* [43]. 

Furthermore, MeJA induced lipids and fatty acids production in several microalgae [28,44,45], and it is known that it specifically upregulates ∆6*Des* gene expression, increasing γ-linolenic acid (GLA) production [46]. NaHCO_3_ also increases the lipids production in microalgae and is considered a cheaper carbon supply than CO_2_ [47,48,49]. Our results show that self-exDNA is a better inductor of lipids production than MeJA and NaHCO_3_ at 24 and 48 h after treatments (Figure 3), but a comparison with nitrogen starvation should be carried out. It is known that MeJA is a good inducer of polyunsaturated fatty acids (PUFA) by increasing the expression of the desaturase gene [46]. For triacylglycerols, self-exDNA is as good an inducer as MeJA and the effect in both cases is apparently very fast since the number of unsaturated compounds is high from 0 h (Figure 4). The effect of nonself-exDNA is interesting because it seems to inhibit the production of triacylglycerols but not lipids (Figure 3). Indeed, nonself-exDNA induces lipids production in a similar manner as MeJA and NaHCO_3_ but not as well as self-exDNA. It was reported that oleic acid production increased with the addition of MeJA, and other factors with significant effects on lipid production were NaH_2_PO_4_, GA, and IAA [50]. 

Biosynthetic pathways of phytohormones in microalgae, although rudimentary, share some key components compared to vascular plants. Microalgae phytohormones have a dual function; they control the cell cycle regulation and other metabolic processes such as biomass and primary metabolite accumulation [51]. Additionally, they are involved in responses to abiotic stresses, allowing adaptation to prevailing conditions. Therefore, phytohormones have a role in microalgal growth and survival, increasing lipid productivity and improving tolerance to extreme environmental changes [51]. In general, green microalgae are divided into two lineages, Chlorophyta comprising the majority of species, and Streptophyta comprising the Charophyta which gave rise to land plants [52,53]. Thus, the adaptation to the terrestrial plant habitat has been linked to an expansion of the genome and gene expression [52,54]. Similar metabolic pathways may have occurred in microalgae and in vascular plants since few exclusive genes, receptors, and enzymes related to the biosynthesis and signaling pathways of various phytohormones have been identified in some Chlorophyta and Streptophyta [53,55,56]. The analysis of axenic microalgal strains provided convincing evidence that they can synthesize phytohormones. 

It has been speculated that auxins, and particularly IAA, are synthesized via the indole-3-acetamide (IAM) pathway, as well as possibly the tryptamine (TRA) pathway, based on the reported microalgae auxin profiles [57]. We detected an average of 100 pM of IAA in *N. oleoabundans*, but any treatment had an effect on the IAA production (Figure 5a). 

N^6^-substituted adenine derivatives are known as cytokinins and can be classified by their side chains as either isoprenoid or aromatic. Cytokinins are synthesized from dimethylallyl pyrophosphate to ribotides, the immediate cytokinin precursors. Two pathways are involved, the mevalonate (MVA) and the methyl-erythritol-phosphate (MEP) by reactions catalyzed by isopentenyltransferases [58]. The four groups of isoprenoid cytokinins are *trans*-zeatin (*t*Z), *cis*-zeatin (*c*Z), dihydrozeatin (DHZ), and isopentenyladenine (IsoA); 6-benzylaminopurine (BAP) is also considered a cytokinin. Based on the cytokinin profiles, it seems that a simplified cytokinin network exists in microalgae compared to vascular plants. Degradation of tRNA is also a likely source of cytokinins in microalgae [58]. We detected an average of 55 pM of IsoA and 65 pM of BAP average in *N. oleoabundans*, interestingly self-exDNA induced the production of both cytokinins after 24 h of treatment. MeJA also induced the IsoA production in a similar manner to self-exDNA (Figure 5b,c).

Gibberellins are tetracyclic diterpene carboxylic acids synthesized from geranylgeranyl diphosphate, a common precursor for diterpene and carotenoid compounds. Between 16 and 20 gibberellins have been detected in 24 microalgae species [51]. According to the gibberellin profiles, it looks like the biosynthesis goes by the hydroxylated pathway to produce GA_6_, with the oxidation pathway to form GA_4_ being a minor pathway [59]. In *N. oleoabundans*, we detected 27 pM averages of GA from 0 h, except with MeJA where no GA was detected that time. After 24 h, GA production was significantly induced by self-exDNA and MeJA (Figure 5d). 

In plants, jasmonates are synthesized in the chloroplasts and peroxisomes by the octadecanoid pathway. Under certain conditions, α-linolenic acid is released from the membrane’s lipids and is oxidized to 12-*oxo*-phytodienoic acid, which is converted by reduction and oxidation reactions to methyl jasmonate (MeJA). In the genomes of microalgae, two putative JA biosynthetic orthologs have been identified, suggesting that they may synthesize jasmonates [60]. In this work, around 60 pM MeJA was detected in *N. oleoabundans* at 0 h, and any treatment affected its production for 48 h (Figure 5e). The external MeJA added at 0 h was still detected at 24 but not at 48 h, suggesting degradation. It is worth mentioning that self-exDNA did not induce the MeJA production, but both factors showed a similar behavior in inducing peroxidase enzyme activity, polyphenols, lipids, triacylglycerols, GA, IAA, and IsoA production. It was reported that ethanol induced an oxidative response in *Haematococcus pluvialis*, which increased the lipids production and antioxidant enzymes activities, as well as the MeJA biosynthesis [61]. Therefore, if self-exDNA induced an oxidative environment into the *N. oleoabundans* cell, then further investigation should be conducted to determine why it did not induce the MeJA biosynthesis as expected. Further work needs to be done to explain the fact that self-exDNA fails to induce MeJA biosynthesis in *N. oleoabundans*. 

In plants, SA is a key factor to regulate stress responses, it activates antioxidant enzyme activities enhancing the antioxidant capacity and inducing genes responsible for protective mechanisms [62]. In plants, the phenolic compound SA is derived from the shikimate–phenylpropanoid pathway where phenylalanine is converted to cinnamic acid. Then, this is hydroxylated to form *o*-coumaric acid, and the side-chain is oxidized to form benzoic acid (BA), which is then further hydroxylated to produce the SA [62]. SA has been detected in *C. minutissima* and *Scenedesmus quadricauda* [63,64]. Recently, BA and SA have been proposed as signaling molecules regulating cell-to-cell communication in *Chlorella regularis*. A CoA-dependent, non-oxidative pathway from *trans*-cinnamic acid has been proposed for the biosynthesis of BA and SA in these microalgae [65]. We detected about 250 pM SA in all samples at 0 h; however, SA decreased to 200 pM in all treatments at 24 h except for control samples (Figure 5f). At 48 h, all samples had about the same SA average concentration of 200 pM. Although this effect of diminution of SA concentration in treated samples was not statistically significant, it is possible that the biosynthetic SA pathway could be inhibited in all treatments in *N. oleoabundans*.

Most of the studies on microalgae have been carried out with the intention of overproducing valuable and useful products such as lipids, carbohydrates, and others. However, there is still a deep lack of study of the basic metabolism of microalgae in general. The fact that self-exDNA produces an oxidative status on *N. oleoabundans*, similar to the effect on plants [66], and that these effects are so specific suggests that defensive mechanisms are being used to avoid the introduction of potentially harmful material. It is currently unknown if microalgae have pattern recognition receptors (PRRs) and potential mechanistic scenarios for external DNA and RNA perception as those described in plants [67], which are a key part of the immune system in plants and animals, and a sophisticated mechanism to respond to biotic stresses. Those PRRs, which are cell-surface-localized receptor kinases (RKs) or receptor proteins (RPs), are involved in sensing microbe- or self-derived molecular patterns to regulate pattern-triggered immunity (PTI), a form of antimicrobial immunity [9]. Recognized elicitors of those PRRs are Microbe- or Pathogen-Associated Molecular Patterns (MAMPs or PAMPs), and endogenous Damage-Associated Molecular Patterns (DAMPs) or peptides which are processed and/or secreted upon infection known as phytocytokines [6,68,69,70,71]. Moreover, we cannot exclude the possibility that *N. oleoabundans* and other microalgae may produce an extracellular matrix (ECM) similar to neutrophils extracellular traps (NETs) described in animals, root extracellular traps (RETs) in plants or structures similar to bacterial biofilms, having external self-exDNA playing an active role [72]. Therefore, according to the similitudes among microalgae and plants regarding defensive mechanisms, it is expected that microalgae also have PRRs and/or ECM. Our results strongly suggest that these might be present in *N. oleoabundans*.

There are still many questions to answer such as what the molecular traits giving such specificity to recognize the self-exDNA over others are, and if defense responses elicited by self-exDNA are really similar to those induced by DAMPS, or if there could be some differences. The oxidative environment induced by self-exDNA in *N. oleoabundans* seems to be different from that induced by ethanol, because the latter induces MeJA biosynthesis in *H. pluvialis* [61], but self-exDNA did not induce MeJA in *N. oleoabundans*. Unless deep metabolic differences could exist between *H. pluvialis* and *N. oleoabundans*, which does not seem to be feasible, it is possible that different “oxidative environments” could be induced by biotic and abiotic elicitors. In addition, the induction of lipids biosynthesis by self-exDNA in *N. oleoabundans* needs to be compared to the same effect but induced by N starvation [21,73]. We do not know if both treatments could be synergistic or antagonistic, or even if the action mechanism could be different, although the final result would be the increase in lipids production. The timing of responses and the modulation of the effect at different dosages of both self- and nonself-exDNA, needs to be addressed. We do not know if RNAs may play some role in this induction, but we know that nucleotides have an elicitor effect [15]. In the case of transgenic organisms, it will be interesting to know if the introduced foreign DNA could be recognized as self by the original organism, which may have serious consequences. 

## 4. Materials and Methods

### 4.1. Strain and Crops

The microalgae *N. oleoabundans* UTEX 1185 were obtained from the University of Texas at Austin, TX, USA and grown under axenic conditions. Cultures were in 500 mL bioreactors (Pyrex^®^ 1395-500, Corning, NY, USA.) with Bold Basal culture medium [74], in a controlled 12 h light/dark photoperiod with fluorescent cold white light at 50 μmol photons m-2s-1 (PAR) at 25 °C. Ambient air filtered through a 0.2 μm pore venting filter (Millex^®^-FG 50 Millipore SLFG05010) was added at 1 VVM. At different times, cells were counted with a Neubauer chamber, and growing was followed by absorbance at 750 nm.

### 4.2. Microalgal DNA Extraction

When *N. oleoabundans* cell culture measured a DO 0.4 at 750 nm in a Multiskan Go spectrophotometer (Thermo scientific, San Jose, CA, USA), it was centrifuged at 5000× *g* in a Sorvall 3000 centrifuge. Cells were grinded with liquid nitrogen in 0.5 mL lysis buffer (1% SDS, 50 mM EDTA and 100 mM Tris) and vortexed 4 times 2 min each. A measure of 1 mL of DNAse inhibitor (Invitrogen, San Jose, CA, USA) at 1 mg mL^−1^ was added and incubated for 10 min at 37 °C. Then, 10 µL proteinase K (Invitrogen, San Jose, CA, USA) to 0.1 mg mL^−1^ was added and incubated for 60 min at 60 °C. Proteins were eliminated with same volume of phenol: chloroform: isoamyl alcohol (25:24:1), and after being vortexed, the mixture was centrifuged at 13,500× *g* for 15 min. Upper aqueous phase was recovered and added with 500 μL chloroform, vortexed, and centrifuged at the same speed for 5 min. The upper aqueous phase was transferred to a new tube and added with 500 μL cold isopropanol, mixed by inversion, and settled for 30 min at 4 °C. It was centrifuged at the same speed and time, and the DNA pellet obtained was washed 3 times with 70% ethanol. The pellet was allowed to dry and resuspended in 200 μL sterile deionized water. Nucleic acids quantification was in a UV–Vis Quawell NanoDrop spectrophotometer. DNA integrity was assessed by electrophoresis in 1% agarose gel.

### 4.3. DNA Fragmentation

Self and nonself-exDNA from salmon sperm (S3126-1G type IIS from Sigma-Aldrich, St. Louis, MO, USA) were sonicated using a CV18 Ultrasonic processor. Each DNA solution was busted for 8 min at 60 s intervals, with 20% amplitude and a control of 10% pulses per cycle. Both were sonicated as described until having 250–2000 bp fragments. Fragmentation was confirmed by agarose gel electrophoresis with 1 Kb molecular marker.

### 4.4. Treatments

For all treatments, algal culture was used at OD_750_ = 0.435 from a six-day culture. At that point, the culture had an average of 482 million cells mL^−1^. A dose–response assay of the peroxidase enzyme activity of *Neochloris oleoabundans* was carried out, after the application of self-exDNA at different concentrations (0, 5, 10, 20, and 30 ng μL^−1^) and times (0, 15, 30, 60, and 120 min). This was done to define the optimal concentration of self-exDNA at which the first statistically significant response was observed. The resulting data were plotted as shown in Appendix A. According to these results, *N. oleoabundans* inductions were performed with self-ex and nonself-exDNA at 20 ng μL^−1^. Phytohormone methyl jasmonate (95% Sigma-Aldrich) was used at 10 µM, and NaHCO_3_ (KARAL) at 120 mM. All inductions were performed in triplicate.

### 4.5. Peroxidase Enzyme Activity Assay

Enzyme activity was measured spectrophotometrically at 480 nm using guaiacol and hydrogen peroxide as hydrogen donor and substrate, respectively [75,76,77], with some modifications. A measure of 5 mL *N. oleoabundans* culture was treated with each elicitor and samples taken at 0, 5, 10, 15, 30, 60, and 120 min after induction. Samples were centrifuged and each pellet was resuspended in 200 μL 0.1 M phosphate-buffered saline (PBS, 137 mM NaCl, 2.7 mM KCl, 10 mM Na_2_HPO_4_, and 1.8 mM KH_2_PO_4_); all next steps were ice cooled. Samples were centrifuged again and after discarding the supernatant, 80 μL PBS + 20 μL 0.1% Triton were added and sonicated for 4 cycles of 60 s of sonication and 60 s rest. Cell extract was centrifuged, and supernatant was used to carry out the enzyme assay. In a 96-well plate, we added 50 μL 0.1 M PBS, 50 μL cell extract, 50 μL 5 mM guaiacol, and 50 μL H_2_O_2_. The plate included a control without cell extract and a positive control containing peroxidase enzyme (Sigma-Aldrich, St. Louis, MO, USA). The extinction coefficient of guaiacol (ε = 26.6 mM cm^−1^) was used to calculate the enzymatic activity.

### 4.6. Determination of Total Polyphenols

Samples of 5 mL were centrifuged at 10,000× *g* for 15 min at 4 °C, the supernatant was discarded, and the obtained biomass was lyophilized. Then, polyphenols were extracted with 250 μL 70% methanol for 12 h at room temperature under dark conditions in a rotary shaker at 250× *g*. Samples were centrifuged at 10,000× *g* for 15 min and supernatants were obtained and assayed for polyphenols. The assay was as reported [78] adapted to a microplate format. Briefly, 237 μL sterile distilled water and 3 μL crude extract were mixed with 15 μL Folin–Ciocâlteu reagent diluted in water (1:1). Then, 45 μL 1 M sodium carbonate was added, and the mixture was incubated for 30 min at 25 °C in the dark. Absorbance was measured at 760 nm in a Multiskan Go spectrophotometer (Thermo scientific), and total phenolic content was expressed as gallic acid equivalents (GAE) in milligrams per sample gram. A standard curve of gallic acid was previously developed.

### 4.7. Gravimetric Determination of Total Lipids

Total lipids were extracted from 30 mL *N. oleoabundans* samples, after centrifugation at 10,000× *g* by 15 min at 4 °C. Pellets were extracted with MeOH:CH_2_Cl_2_ (1:1) added with butylhydroxytoluene (BHT) 0.05% to avoid lipid oxidation, by at least 12 h at 4 °C. Cells were centrifuged as previously described and the pellet was washed twice by vigorously shaking with 4 mL MeOH:CH_2_Cl_2_, mixed, and centrifuged. Supernatants were mixed and the same final volume of H_2_O was added. The lower organic phases containing the lipids in CH_2_Cl_2_ were separated and the remaining water was extracted with Na_2_SO_4_ anhydrous and filtered on Whatman No. 1 paper. Organic phases were concentrated in a rotary evaporator at 45 °C and the extracted lipids were placed in pre-weighted vials. Samples were dried with nitrogen gas flow until constant weights were obtained. Total lipid content was determined by weight difference. Then, 1 mL of crude oil from *Neochloris oleoabundans* was placed in a separatory funnel and 20 mL of methanol (20:1) was added at room temperature and shaken vigorously, 20 mL of hexane was added and shaken again vigorously. To break the equilibrium, 20% of water with respect to methanol (4 mL) was added and shaken for the last time. Two phases were formed, the upper part of hexane containing the lipids and a lower phase of methanol and water. The lower phase was decanted, and the hexane phase was recovered, and concentrated in a rotary evaporator at 40 °C, the obtained oil was resuspended in heptane. Then, it was applied to a column (50 cm long × 1.6 cm diameter) packed with 40 g of 60 G silica gel (Merck) suspended in 150 mL heptane. The next solvents were added one by one in the following order: (1) 50 mL heptane, (2) 50 mL benzene, (3) 50 mL chloroform, (4) 50 mL chloroform:methanol (2:1), and (5) 50 mL methanol. The first yellow fractions were collected and used for HPTLC, and the green-orange fractions were discarded. The column was regenerated with several washes with heptane.

### 4.8. Qualitative Analysis of Triacylglycerols by HPTLC

The method of the International Association for the Advancement of High Performance Thin-Layer Chromatography was used (HPTLC Association https://www.hptlc-association.org/about.cfm. Accessed on 24 November 2022) with minor modifications. Samples of 1 mg of *N. oleoabundans* oil and standards were dissolved in 1 mL of chloroform–phenol (2:1). Standards were triolein, which is a symmetrical triglyceride derived from glycerol and three units of oleic acid (C18:1); trilinolein, with glycerol and three units of linoleic acid (C18:2); and trilinolenin, with glycerol and three units of linolenic acid (C18:3) [25]. A silica plate Si 60 reversed phase (RP)-18 F254 (Merck, Tokyo, Japan) was used with a CAMAG TLC SAMPLER 4 automatic applicator, where 8 mm long and 10 mm apart bands were placed. Dichloromethane, acetic acid, and acetone 20:40:50 (*v*/*v*/*v*) were used as mobile phase and developed in a CAMAG DVELOPING CHAMBER 2 (ADC 2) automatic development chamber, saturated with the mobile phase. The development distance was 80 mm from the bottom edge, and the relative humidity of the chamber was 33%. Derivatization was performed by spraying with 5 g phosphomolybdic acid dissolved in 200 mL of ethanol and heating the plate at 100 °C for 2 min on a CAMAG TLC/HPTLC Plate Heater 3. The plate was visualized and documented in a professional CAMAG viewer for white light chromatograms in remission and transmittance (RT). Image analysis was performed with the visionCATS program (https://www.camag.com/products/software#downloads. Accessed on 24 November 2022). 

### 4.9. Extraction, Purification, and UHPLC-ESI-MS/MS Analysis of Phytohormones

After induction times, samples of 50 mL were centrifuged at 5000× *g* and 4 °C, the biomasses were lyophilized. Phytohormones extraction were performed according to the reported method with minor modifications [79]. Lyophilized samples were suspended in 1 mL of extraction solvent (acetonitrile:H_2_O, 1:1), homogenized for 5 min with a high-speed homogenizer (Ultraturrax T10, IKA, Staufen, Germany) at 4 °C, and sonicated for 3 min at 4 °C with an ultrasonic bath (Branson 2800, Branson Ultrasonic Corporation, Danbury, CT, USA). Phytohormones were extracted for 30 min at 4 °C with occasional rotation, then samples were centrifuged for 20 min at 16,000× *g* at 4 °C. Supernatants were purified using 100 mg solid phase extraction cartridges (Hypersep C18, Thermo Scientific, San Jose, CA, USA). Cartridges were preconditioned with 1 mL methanol and washed with 1 mL H_2_O and 1 mL extraction solvent, then samples were applied, and the flow channel collected. Cartridges were washed with 1 mL 30% acetonitrile and eluates were collected. Organic solvents were removed by SpeedVac, and the polar solvent was removed by lyophilization; dried samples were stored at −20 °C until analysis. Samples were suspended in 100 µL 30% acetonitrile and transferred to 300 µL glass insert vials. Chromatographic analysis was performed on a UHPLC Ultimate 3000 (Thermo Scientific, San Jose, CA, USA) coupled with an Orbitrap Velos Pro mass spectrometer (Thermo Scientific, San Jose, CA, USA). Chromatographic separation was performed on a Hypersil GOLD C18 reverse phase column (1.9 µm particle size, 100 mm× 2.1 mm) at 50 °C. Phytohormones were eluted with a binary gradient of water (A) and methanol (B), both with 0.1% acetic acid at a flow rate of 0.3 mL min^−1^. The applied gradient program was 5% B for 5 min, 5–100% B for 17 min, and hold for 5 min. All samples were recorded at 210 nm in the UV–Vis module. UHPLC-ESI-MS/MS (SRM) was used for the quantification of *t*Z/*c*Z, IsoA, BAP, IAA, MeJA, SA, ABA, and GA3 in a range between 45 and 570 picomoles (pM). ESI source conditions were a spray voltage of 4.5 kV, capillary temperature of 300 °C, heater temperature of 180 °C, sheath gas flow rate of 40 arbitrary units, and auxiliary gas of 15 arbitrary units. Data were processed using Qualbrowser and Quanbrowser implemented in Xcalibur version 4.1 (Thermo Fisher Scientific, Waltham, MA, USA). The limit of detection (LOD) and limit of quantification (LOQ) were calculated as $\frac{3.3S_{x/y}}{b}$ and $\frac{10S_{x/y}}{b}$, respectively. The LOD and LOQ for each phytohormone were determined reflecting their physicochemical properties and ionization behavior. In the UHPLC-ESI-MS/MS analysis of phytohormones, transitions obtained for each analyte by SRM are shown in Appendix A. To validate the method, standards containing the phytohormones of interest were analyzed and calibration curves were established for each analyte based on the areas of serially diluted authentic standard solutions ranging from 15.625 to 500 pg. The resulting data were converted to pM and plotted as shown in Appendix A. All plots were linear over at least seven points on the calibration curves with R^2^ values ≥ 0.999 (Appendix A).

### 4.10. Statistical Analysis

The R-Studio program was used for statistical analysis using the one-factor analysis of variance (ANOVA) method to examine the differences in the means of three or more groups, applying the Shapiro–Wilk normality test applicable in composite samples of at least 50 elements. If necessary, the ANOVA or Welch methods were used to compare two means, or the non-parametric alternative of the ANOVA method, the Kruskall–Wallis test, was used to check the hypothesis that k quantitative samples were obtained from the same population.

## 5. Conclusions

This study was conducted to analyze the response to DAMPs, especially to self-exDNA in the microalgae *N. oleoabundans*. We found that the microalgae respond as other living organism to DAMPs with the activation of ROS, polyphenol content to alleviate the oxidative burst, and specifically as plants with phytohormonal activation. The response was more evident and stronger with their self-exDNA compared with a very distant phylogenetic nonself-exDNA (salmon sperm DNA). As far as we know, this is the first evidence that self-exDNA induces metabolic changes in microalgae.

## Figures and Tables

**Figure 1 ijms-24-14172-f001:**
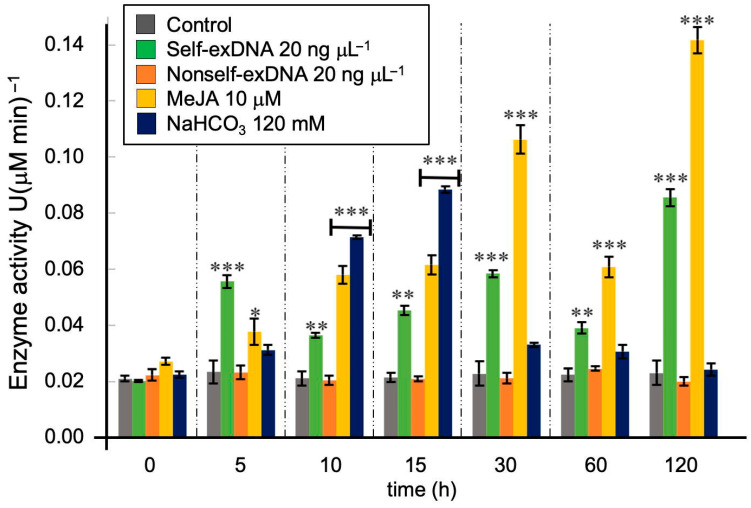
Effects of elicitors on peroxidase enzyme activity of *Neochloris oleoabundans*. The unit of enzyme activity (U) was defined as the catalytic activity responsible for the transformation of 1 µmol substrate per minute under optimal enzyme conditions. Data represent mean ± standard deviation (SD, n = 3). Asterisks indicate statistically significant differences (* = *p* < 0.05; ** = *p* < 0.01; *** = *p* < 0.001).

**Figure 2 ijms-24-14172-f002:**
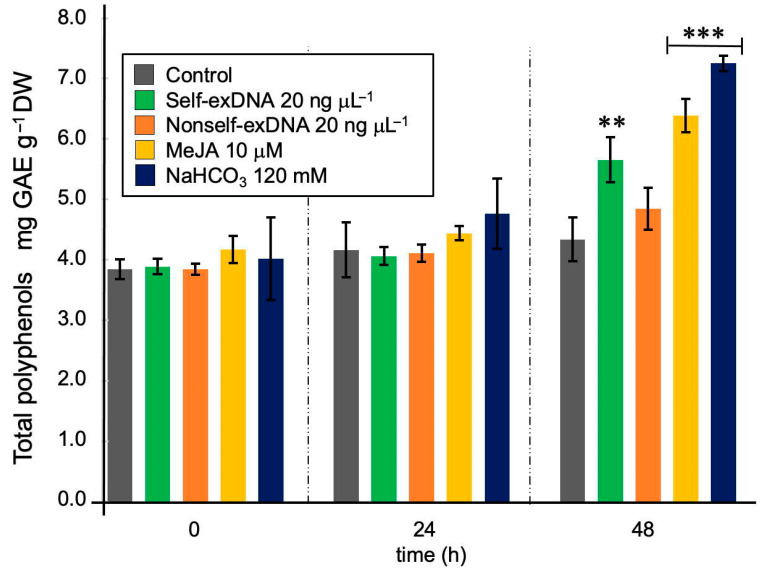
Effects of elicitors on the production of total polyphenols in *Neochloris oleoabundans*. Total phenolic content was expressed as gallic acid equivalents (GAE) in milligrams per sample gram dry weight. Data represent mean ± standard deviation (SD, n = 3). Asterisks indicate statistically significant differences (** = *p* < 0.01; *** = *p* < 0.001).

**Figure 3 ijms-24-14172-f003:**
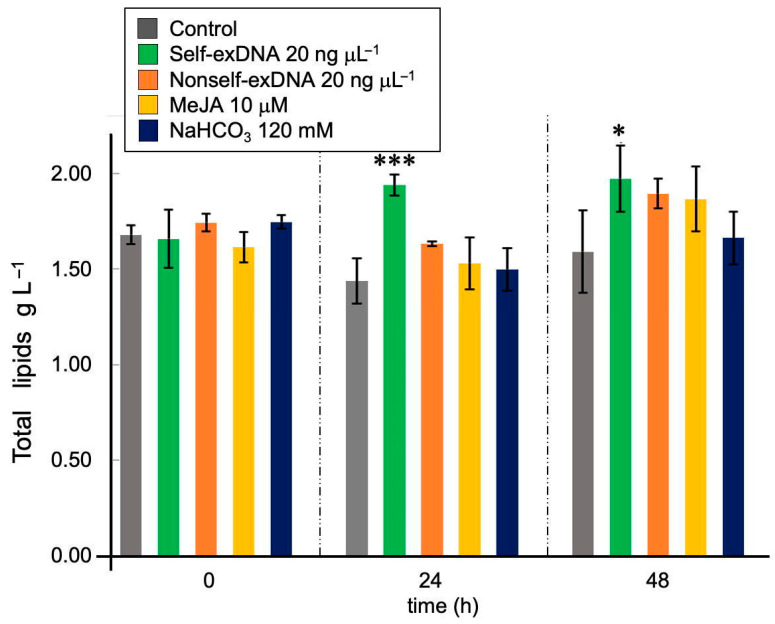
Effects of elicitors on the total lipid production in *Neochloris oleoabundans*. Data represent mean ± standard deviation (SD, n = 3). Asterisks indicate statistically significant differences (* = *p* < 0.05; *** = *p* < 0.001).

**Figure 4 ijms-24-14172-f004:**
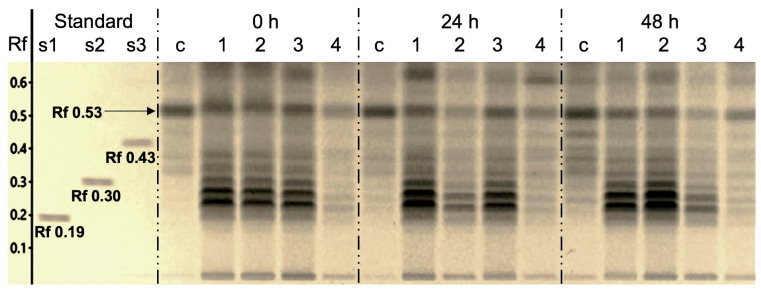
Chromatographic profile of *N. oleoabundans* triglycerides in HPTLC. (Rf), Retention factor; Standards: (s1), trilinolenin (3C18:3); (s2), trilinolein (3C18:2); (s3), triolein, (3C18:1); (c), control; (1), MeJA 10 mM; (2), NaHCO_3_ 120 mM; (3), self-exDNA 20 ng μL^−1^; (4), nonself-exDNA 20 ng μL^−1^.

**Figure 5 ijms-24-14172-f005:**
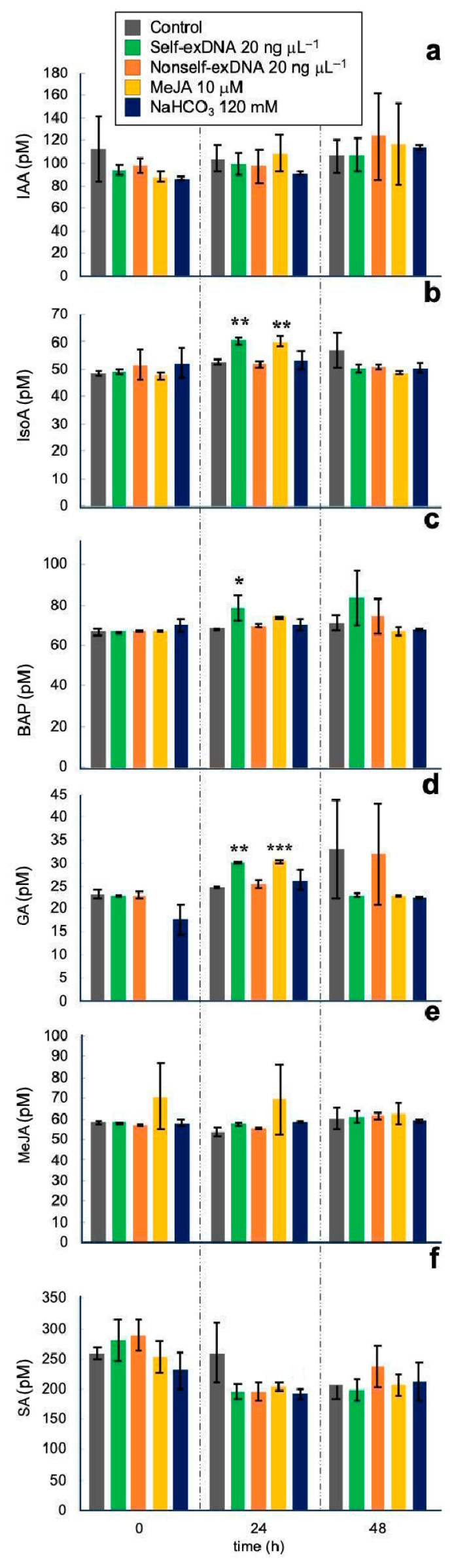
Effect of the elicitors on the production of phytohormones in *Neochloris oleoabundans*. (**a**), IAA; (**b**), IsoA; (**c**), BAP; (**d**), GA; (**e**), MeJA; (**f**), SA. Data represent mean ± standard deviation (SD, n = 3). Asterisks indicate statistically significant differences (* = *p* < 0.05; ** = *p* < 0.01; *** = *p* < 0.001).

## Data Availability

Original data can be made available upon request.

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
