# Peer review of "Metabolic Responses of the Microalga Neochloris oleoabundans to Extracellular Self- and Nonself-DNA"

_ijms, 2023, doi:10.3390/ijms241814172_

Round 1

Reviewer 1 Report

The paper titled "Metabolic responses of the microalga Neochloris oleoabundans to extracellular self- and nonself-DNA" is well structured and clearly presented. I find some minor corrections to be done that are listed below:

- lines 130, 143, 155, 177 e 179: “N. oleoabundans” must be in italic

- lines 229-232 remove italic

- line 306 "that self-exDNA fail to ..." --> fails to

- line 308: remove double dot

The discussion ends in a rather abrupt way. I suggest adding some final considerations on possible further research directions and possible other topics and organisms of interest.

The discussion could also include a sentence suggesting further studies on the molecular synthesis of the investigated compounds to better understand the timing of responses and the modulation of the effect at different dosage of both self and nonself DNA.

The Conclusion paragraph after the Methods at the end of the paper reads a bit disconnected to the main body of the paper. Consider anticipating it after the Discussion.

At line 325, when mentioning the effects on plants a proper reference to be indicated could be Chiusano et al. (2021). Arabidopsis thaliana response to extracellular DNA: self versus nonself exposure. Plants 10.8: 1744.

Regarding the exDNA receptors mentioned at lines 327-334 two additional useful references could be:

Bhat and Ryu (2016). Plant perceptions of extracellular DNA and RNA. Mol. Plant 9, 956–958. doi: 10.1016/j.molp.2016.05.014

Monticolo et al. (2020). The role of DNA in the extracellular environment: a focus on NETs, RETs and biofilms. Frontiers Plant Science, 11, 589837.

Author Response

Comments from Reviewer 1 and Answers (in Italics and BOLD)

The paper titled "Metabolic responses of the microalga Neochloris oleoabundans to extracellular self- and nonself-DNA" is well structured and clearly presented. I find some minor corrections to be done that are listed below:

- lines 130, 143, 155, 177 e 179: “N. oleoabundans” must be in italic   Done

- lines 229-232 remove italic                                                                   Done

- line 305 "that self-exDNA fail to ..." --> fails to                                   Done

- line 308: remove double dot                                                                  Done

The discussion ends in a rather abrupt way. I suggest adding some final considerations on possible further research directions and possible other topics and organisms of interest. The discussion could also include a sentence suggesting further studies on the molecular synthesis of the investigated compounds to better understand the timing of responses and the modulation of the effect at different dosage of both self and nonself DNA. In response to this comment, lines 342 – 358 were added as follows: “There are still many questions to answer like what’s the molecular trait(s) giving such specificity to recognize the self-exDNA over others, and if defense responses elicited by self-exDNA, are really similar to those induced by DAMPS, or if there could be some differences. The oxidative environment induced by self-exDNA in N. oleoabundans, seems to be different form that induced by ethanol, because last one induces MeJA biosynthesis in H. pluvialis [61], but self-exDNA didn’t induce MeJA in N. oleoabundans. Unless deep metabolic differences could exist between H. pluvialis and N. oleoabundans, which doesn’t seem to be feasible, it’s possible that different oxidative environments could be induced by biotic and abiotic elicitors. Also, the induction of lipids biosynthesis by self-exDNA in N. oleoabundans needs to be compared to the same effect but induced by N starvation [79]. We don’t know if both treatments could be synergistic or antagonistic, or even if the action mechanism could be different although the final result would be the increase in lipids production. The timing of responses and the modulation of the effect at different dosage of both self- and nonself-exDNA, needs to be addressed. We don’t know if RNAs may play some role in this induction, but we know that nucleotides have an elicitor effect [15]. In the case of transgenic organisms, it will be interesting to know if the introduced foreign DNA could be recognized as self by the original organism, which may have serious consequences”. We hope this addition will be accepted by the Reviewer 1.

The Conclusion paragraph after the Methods at the end of the paper reads a bit disconnected to the main body of the paper. Consider anticipating it after the Discussion.                                                                             We agree with the Reviewer 1; however, in the IJMS Word Template provided from same journal, Conclusions are placed in the 5th position, after Material and Methods. We just followed the format, and we think the final decision will be from the journal.

At line 325, when mentioning the effects on plants a proper reference to be indicated could be: Chiusano, M. L., Incerti, G., Colantuono, C., Termolino, P., Palomba, E., Monticolo, F., Benvenuto, G., Foscari, A., Esposito, A., Marti, L., de Lorenzo, G., Vega-Muñoz, I., Heil, M., Carteni, F., Bonanomi, G. & Mazzoleni, S. (2021). Arabidopsis thalianaresponse to extracellular DNA: self versus nonself exposure. Plants, 10, 1744. https://doi.org/10.3390/plants10081744       In response to this comment, we included it at the end of line 324 as reference [66].

Regarding the exDNA receptors mentioned at lines 327-334 two additional useful references could be: Bhat, A., & Ryu, C. M. (2016). Plant perceptions of extracellular DNA and RNA. Molecular plant, 9, 956-958. http://dx.doi.org/10.1016/j.molp.2016.05.014         In response to this comment, we included it in lines 327-328 as reference [67] with the next addition: “and potential mechanistic scenarios for external DNA and RNA perception as those described in plants [67]”.

Monticolo, F., Palomba, E., Termolino, P., Chiaiese, P., De Alteriis, E., Mazzoleni, S., & Chiusano, M. L. (2020). The role of DNA in the extracellular environment: a focus on NETs, RETs and biofilms. Frontiers in Plant Science, 11, 589837. https://doi.org/10.3389/fpls.2020.589837                                                                                                                  Included in the lines 335-339 as reference [72] with the next addition “Also, we can’t exclude the possibility that N. oleoabundans and other microalgae, may produce an extracellular matrix (ECM) similar to neutrophils extracellular traps (NETs) described in animals, root extracellular traps (RETs) in plants or structures similar to bacterial biofilms, having external self-exDNA playing an active role [72]”.

We thank the Reviewer 1 for the corrections and suggestions which improved the quality of the manuscript.

Reviewer 2 Report

The researchers conducted a study on the stress response of Neochloris oleoabundans under various conditions and aimed to identify self-exDNA as a specific elicitor. However, to arrive at a convincing conclusion, further controlled experiments are required. For instance, the current study utilized only one concentration of self-exDNA and nonself-exDNA (20ng/uL). Therefore, it is essential to establish a dose-response curve. The manuscript contains several typos and requires further proofreading.

The manuscript contains several typos and requires further proofreading.

Author Response

Comments from Reviewer 2 and Answers (in Italics and BOLD)

The researchers conducted a study on the stress response of Neochloris oleoabundans under various conditions and aimed to identify self-exDNA as a specific elicitor. However, to arrive at a convincing conclusion, further controlled experiments are required. For instance, the current study utilized only one concentration of self-exDNA and nonself-exDNA (20ng/mL). Therefore, it is essential to establish a dose-response curve.                                                                                                                  We agree with the Reviewer 2 about this fact and the explanation to use that concentration comes from the results of a previous publication numbered as [17] (Palomba, E., Chiaiese, P., Termolino, P., Paparo, R., Filippone, E., Mazzoleni, S. & Chiusano, M. A. (2022). Effects of Extracellular Self- and Nonself-DNA on the Freshwater Microalga Chlamydomonas reinhardtii and on the Marine Microalga Nannochloropsis gaditana, Plants, 11, 1436. https://doi.org/10.3390/plants11111436 ). This is the only previous publication of the effect of self-exDNA in microalgae and in that paper, the effect of 3, 10 and 30 ng mL-1 of self-exDNA applied to C. reinhardtii and N. gaditana were studied and the 30 ng mL-1 concentration showed growth inhibition in both microalgae since 24 h of treatment. Besides, after self-exDNA application of 30 ng mL-1, both microalgae showed an altered morphology, appearing in a similar palmelloid phenotype and forming aggregates. In comparison, at concentrations of 3 and 10 ng mL-1 , both algae continued growing suggesting an active metabolic condition. So, considering these results [17] and being N. oleoabundans a microalgae, we decided to use 20 ng mL-1and after checking that we had fast and specific results of the antioxidant response of N. oleoabundans as shown in Figure 1, we decided to continue with the rest of the experiments at the same concentration of 20 ng mL-1 for the nonself- and self-exDNA to have a wider view of the response of N. oleoabundans, regarding polyphenols, lipids, triglycerides and phytohormones production. On the other hand, extraction of sufficient pure DNA from microalgae is difficult and time-consuming because it requires many liters of culture. So, in view of the obtained results we decided to keep the 20 ng mL-1 of nonself- and self-exDNA concentrations for all the experiments described in our manuscript. Currently, we are planning other experiments with different nonself- and self-exDNA concentrations mixing those in different proportions but keeping the total amount of DNA. This will allow us to check the effects at a range of concentrations to know when effects starts and the result of higher amounts of self-exDNA, in N. oleoabundans. Considering the mentioned previous results of [17], we may expect the formation of aggregates and growth inhibition around 30 ng mL-1, but this need to be properly confirmed.

The manuscript contains several typos and requires further proofreading.

Comments on the Quality of English Language

       The manuscript contains several typos and requires further proofreading. We agree with the Reviewer 2 and the manuscript was checked by an American colleague which corrected the original version. All corrections are highlighted as requested. We consider that the English language of the revised version is really improved thanks to the recommendation of the Reviewer 2.

We thank the reviewer 2 for the opportunity to explain the reason to use the mentioned concentration and to encourage us to review and correct the English language. That improved the quality¡ty of the document.

Round 2

Reviewer 2 Report

With an emphasis on originality and novelty, this study aims to investigate the effects of a new substrate on an unstudied species, Neochloris oleoabundans. Therefore, the absence of one prior research does not negate the need to construct a dose curve. The omission of such an investigation could potentially undermine the scientific rigor of the manuscript, impacting its suitability for publication in IJMS.

Author Response

We thanks to the Reviewer 2 for the wise and necessary comment about the need of a dose curve. It was clear that the manuscript needed those data to present rigorous and strictly scientific results, supported by statistical data. In our enthusiasm for the interesting, obtained results; we omitted this very important and essential data which gives structure and strength to the whole work. Although we did a preliminary assay to get this data, it was not complete and needed a statistical analysis. So, a new dose-response curve of peroxidase enzyme activity was done, with different concentrations of self-exDNA (0, 5, 10, 20 and 30 ng μL-1) and measured and several times (0, 15, 30, 60 and 120 min). This confirmed that the concentration of 20 ng μL -1 was the optimal for this work as described below. With this addition, we think that the manuscript is now complete, and we hope this can fulfill the requirements of the Reviewer.

Specifically, we make the next additions in the indicated lines as shown in the file below.

2. Results
2.1. Assay of the N. oleabundans peroxidase enzyme activity after treatments.
Lines 111-117
The dose-response curve of the peroxidase enzyme activity of Neochloris oleoabundans, indicated that 20 ng μL-1 was the optimal concentration of self-exDNA at which the first statistically significant response was observed (SFigure 1). As shown in this figure, after 15 min of self-exDNA application, the peroxidase enzyme activity was statistically significant at 20 and 30 ng μL -1. However, at later times (30, 60 and 120 min), the enzyme activity decreased at concentration of 30 ng μL -1, but at 20 ng μL -1 it was still increasing being the highest at 60 min.
3. Discussion
Lines 208-211
The optimal concentration of self-exDNA to get the first significantly response in N. oleoabundans was 20 ng μL-1, similar to that reported [17] where 10 ng μL-1 had no effect in the cell density of Chlamydomonas
reinhardtii and Nannochloropsis gaditana, but 30 ng μL-1 caused a significantly reduced cell density and aggregates in both microalgae.
4. Materials and Methods
4.4. Treatments
Lines 404-409
A dose-response assay of the peroxidase enzyme activity of Neochloris oleoabundans was done, after application of self-exDNA at different concentrations (0, 5, 10, 20 and 30 ng μL-1) and times (0, 15, 30, 60
and 120 min). This was done to define the optimal concentration of self-exDNA at which the first statistically significant response was observed. The resulting data were plotted as shown in SFigure 1
(Supplementary Material).
Supplementary Materials:
Lines 542-549

SFigure 1.
SFigure 1. Dose-response curve of the peroxidase enzyme activity of Neochloris oleoabundans, after application of self-exDNA at different concentrations and times.
The unit of enzyme activity (U) was defined as the catalytic activity responsible for the transformation of one μmol substrate per minute under optimal enzyme conditions. (c), control without self-exDNA. Data represent mean ± standard deviation (SD, n = 3). Asterisks indicate statistically significant differences (* = p < 0.05; ** = p < 0.01; *** = p < 0.001).

The previous SFigure 1 was now named SFigure 2. Also, the name of the colleague which did the dose-response curve, is included into the authors and in the necessary places to give her the corresponding merit for this hard but nice work.

Round 3

Reviewer 2 Report

With the established dose curve and the revised manuscript, I recommend the acceptance of this article for publication. Thank you.